# Live-cell GLUT4 translocation assay reveals *Per3* as a novel regulator of circadian insulin sensitivity in skeletal muscle cells

Rashmi Sivasengh[1,2], Andrew Scott[1] and Brendan M. Gabriel[1,2,*]

## ABSTRACT

Type 2 diabetes (T2D) is a growing global health concern, with skeletal muscle playing a central role due to its contribution to postprandial glucose disposal. Insulin resistance in skeletal muscle often precedes the clinical onset of T2D and is characterised by impaired GLUT4 trafficking. Circadian disruption is increasingly recognised as a contributor to metabolic dysfunction, yet its impact on skeletal muscle insulin sensitivity remains poorly defined. We hypothesised that circadian regulators influence GLUT4 translocation and glucose uptake, contributing to the metabolic impairments observed in T2D. To investigate this, we developed a high-throughput, live-cell GLUT4 translocation assay capable of capturing circadian dynamics in skeletal muscle cells. Using publicly available transcriptomic data from primary human myotubes derived from individuals with and without T2D, our re-analysis identified altered rhythmic expression of several genes, including *PER3*, *ARNTL*, *HOXB5*, and *TSSK6*. Publicly available phenome-wide association study (PheWAS) data further supported associations between these genes and T2D-related traits. Functional validation using siRNA knockdown revealed that *PER3* silencing significantly impaired GLUT4 translocation and glucose uptake in human skeletal muscle cells, while also abolishing rhythmic insulin responsiveness. *ARNTL* knockdown caused a moderate reduction in GLUT4 translocation, suggesting complementary roles in metabolic regulation. Our findings identify *PER3* as a novel circadian regulator of GLUT4 translocation and insulin sensitivity in skeletal muscle. This work also introduces a sensitive, live-cell assay suitable for real-time assessment of GLUT4 dynamics and circadian regulation, offering a powerful platform for discovering new therapeutic targets in T2D.

KEY WORDS: GLUT4 translocation, Skeletal muscle myotubes, Type 2 diabetes, Circadian rhythm, Clock genes

## INTRODUCTION

Type 2 diabetes (T2D) is a growing, global threat to health, placing a high burden on healthcare services. Skeletal muscle is a central organ within the pathology of T2D, being the major storage depot for postprandial glucose disposal (Gabriel and Zierath, 2017). Insulin resistance of skeletal muscle is recognised as a key driver of T2D disease (DeFronzo and Tripathy, 2009). In fact, the changes to metabolic pathways that ultimately lead to insulin resistance of this tissue may be present years before other signs of the development of T2D can be detected (Gabriel and Zierath, 2017). For example, induced pluripotent stem cells from donors with T2D that have been differentiated into myoblasts have multiple defects, including reduced insulin-stimulated glucose uptake and reduced mitochondrial oxidation (Batista et al., 2020). These T2D-associated defects are conserved despite the transformations that these cells undergo during experimental procedures. It has not been fully elucidated how cultured myocytes from T2D donors preserve a dysfunctional phenotype, including insulin resistance (Bouzakri and Zierath, 2007), but this likely reflects genetic background and epigenetic mechanisms.

One key molecular defect that appears to underpin insulin resistance in skeletal muscle is defective trafficking of the glucose transporter GLUT4 to the cell membrane in response to insulin signalling in a temporally appropriate manner. Indeed, insulin resistance impedes the ability of peripheral tissue to respond flexibly to metabolic events throughout the day, and people with T2D have disrupted circadian metabolism (Gabriel and Zierath, 2022). It is known that altered sleep/wake rhythms from shiftwork, sleep disorders, and social jet lag are associated with obesity, T2D, and related disorders (Mokhlesi et al., 2019; Shan et al., 2018; Tan et al., 2018; Vetter et al., 2018), emphasising the role of circadian rhythms in metabolic health. Physiological circadian rhythms are ultimately driven by cell-autonomous circadian rhythms (Panda, 2016). These are generated by a transcription-translation autoregulatory feedback loop composed of transcriptional activators CLOCK and BMAL1 (ARNTL) and their target genes Period (*PER*), Cryptochrome (*CRY*), and REV-ERBα (*NR1D1*), which rhythmically assemble to complete a repressor complex that interacts with CLOCK and BMAL1 to eventually inhibit transcription (Takahashi, 2015). We have shown that cell-autonomous rhythms of *CLOCK* and *BMAL1* are disrupted in skeletal muscle cells from donors with T2D and that this is coupled with a loss of mitochondrial oxidative circadian rhythm (Gabriel et al., 2021). This loss of mitochondrial functional rhythm was linked with a reduction in circadian transcripts associated with the inner-mitochondrial membrane in cells from people with T2D. Further, *in vivo* inner-mitochondrial gene expression was highly associated with whole-body insulin sensitivity when participants underwent a hyperinsulinaemic-euglycaemic clamp and skeletal muscle biopsies (Gabriel et al., 2021; Peek et al., 2013). Additionally, ablation of skeletal muscle clock genes such as *BMAL1* during diet-induced obesity leads to accelerated glucose intolerance and altered glycolytic activity (Chaikin et al., 2025).

Despite evidence linking circadian disruption to impaired glucose metabolism, few studies have directly examined how circadian rhythms influence insulin-stimulated GLUT4 trafficking in skeletal muscle. Existing methods for testing GLUT4 translocation typically rely on endpoint assays that are low throughput, lack temporal

---

[1]The Rowett Institute of Nutrition and Health, School of Medicine, Medical Sciences, Nutrition, University of Aberdeen, Aberdeen AB25 2ZD, UK. [2]Aberdeen Cardiovascular and Diabetes Centre, School of Medicine, Medical Sciences, Nutrition, University of Aberdeen, Aberdeen AB25 2ZD, UK.

*Author for correspondence (brendan.gabriel1@abdn.ac.uk)

R.S., 0009-0006-0044-2740; B.M.G., 0000-0001-6878-8779

*Biology Open*

resolution, and introduce experimental variability (Heckmann et al., 2022), limiting our ability to assess dynamic, circadian regulation of insulin sensitivity. To address this gap, we developed a high-sensitivity, high-throughput, live-cell assay capable of tracking GLUT4 translocation in real time under circadian conditions. Using this novel tool, we hypothesised that loss of rhythmicity in specific circadian genes impairs GLUT4 translocation and glucose uptake, contributing to reduced insulin sensitivity in skeletal muscle. To test this, we reanalysed publicly available transcriptomic datasets from myotubes derived from individuals with and without T2D to identify rhythmically expressed genes disrupted in disease. We then combined phenome-wide association studies (PheWAS) with siRNA knockdown and functional assays to investigate the role of these candidate genes in GLUT4 translocation and glucose uptake. Our findings identify *PER3* as a novel regulator of circadian insulin sensitivity in skeletal muscle and demonstrate the utility of this new live-cell assay for discovering dynamic regulators of metabolic function.

## RESULTS

### Disrupted circadian rhythms in skeletal muscle cells from individuals with T2D

Re-analysis of publicly available data (Gabriel et al., 2021) demonstrated that *ARNTL*, *HOXB5*, *PER3*, and *TSSK6* had loss of rhythmicity or a differential circadian rhythmicity in skeletal muscle cells from people with T2D compared to healthy individuals matched for age and body mass index (BMI) (Fig. 1A). Our analysis of publicly available data revealed that *ARNTL*, *HOXB5*, *PER3*, and *TSSK6* were all significantly associated with T2D across a variety of databases (Table 1). We also analysed these data using MetaCycle. After plotting amplitude (Fig. 1B), we identified that *PER3*, *CLOCK*, *NR1D1* and *ARNTL*, while retaining rhythmicity, displayed significant reductions in amplitude in myocytes from individuals with T2D compared to those with normal glucose tolerance (NGT), with fold changes of 0.26, 0.31, 0.22 and 0.71, respectively.

### Live-cell GLUT4 translocation assay: model validation

To evaluate the utility of our novel system for monitoring circadian insulin sensitivity, we developed stable skeletal muscle cell lines expressing HiBiT-tagged GLUT4 (Fig. S1, representing the plasmid map). Quantification of GLUT4-HiBiT expressions in these clones was performed using the Nano-Glo HiBiT Assay as depicted in Fig. 2A. The assay detected significant luminescence in a subset of clones transfected with 4000 ng DNA, with mean luminescence values exceeding the baseline established by non-transfected controls. The observed luminescence levels were approximately 1000-fold (Fig. 2B) higher than those of controls, confirming specific expression of the HiBiT-tagged GLUT4 protein. We next conducted an insulin dose-response experiment using L6 cells transfected with the GLUT4-HiBiT construct. The resulting data exhibited a linear increase in luminescence with increasing insulin concentration. Cells expressing the GLUT4-HiBiT construct exhibited a substantial increase in luminescence, approximately 120-fold (Fig. 2C) higher compared to that of uninduced cells (0 nM insulin). The luminescent response reached its maximum at an insulin concentration of 40 nM, indicating saturation of the GLUT4 translocation mechanism at this dose. These findings demonstrate that L6 cell-derived myotubes possess a high degree of insulin sensitivity, evidenced by their robust luminescent response. Moreover, the assay displayed excellent throughput and sensitivity, providing an effective means to quantify GLUT4 translocation in skeletal muscle-derived cells.

### Testing circadian dose-dependent GLUT4 translocation response to insulin

To evaluate the impact of different insulin concentration (0 nM to 50 nM) on circadian rhythmicity in gene expression, JTK analysis was conducted on time-course data. The analysis revealed that rhythmicity was significant only at 5 nM and 30 nM insulin concentrations (Fig. 3), with both conditions exhibiting a consistent 24-h period. Other concentrations lacked statistically significant

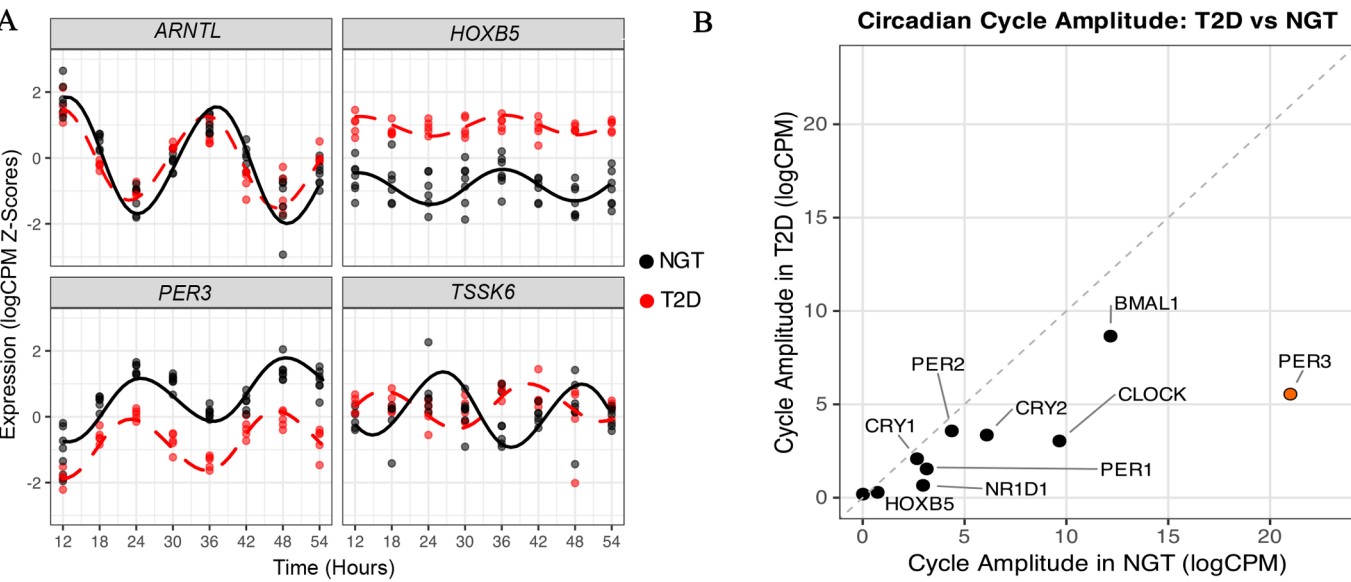

**Fig. 1. Disrupted circadian gene expression in myocytes from people with T2D.** (A) Circadian rhythmicity in gene expression, comparing T2D (shown in red, *n*=5) with NGT (in black, *n*=7). The z-score of logCPM is plotted against time since synchronisation, illustrating gene expression patterns over a 24-h cycle. Harmonic regressions fitted to the data highlight the circadian trends in both groups. The comparison emphasises the distinct circadian gene expression profiles between T2D and NGT individuals. (B) Circadian gene cycle amplitude comparison for selected core clock genes between T2D and NGT for samples from Gabriel et al. (2021) (GEO accession GSE182117). Expression data, as logCPM, were analysed using the JTK_CYCLE (MetaCycle v1.2.0).

Biology Open

**Table 1. Significant genes identified from a public database**

| Gene | Atlas ID | PMID | Year | Domain | Trait | *P*-value | *N* |
|---|---|---|---|---|---|---|---|
| *ARNTL* | 4045 | 30054458 | 2018 | Endocrine | T2D | 0.003 | 659,256 |
| *HOXB5* | 4083 | 28566273 | 2017 | Endocrine | T2D | 0.0001 | 159,208 |
| *PER3* | 4176 | 30718926 | 2019 | Endocrine | T2D | 0.014 | 191,764 |
| *TSSK6* | 4176 | 30718926 | 2019 | Endocrine | T2D | 0.03 | 191,764 |

Query results from https://atlas.ctglab.nl/PheWAS.

rhythms, suggesting a dose-dependent disruption or dampening of circadian control (Table 2). Due to the homogeneity of response and the greater amplitude of rhythmicity, 30 nM insulin, which is close to a physiological insulin concentration, was selected for further experiments.

### Mixed effect modelling to access robust statistical inference of time-dependent translocation

To evaluate the effect of total GLUT4 concentration on GLUT4 translocation efficiency at 30 nM insulin (a concentration at which circadian rhythm effects are significant), surface GLUT4 concentrations (luminescence) were measured over time. Total GLUT4 concentration was determined by lysing cells and measuring luminescence using the HiBiT Lytic Assay Kit (Promega). The resulting surface-to-total GLUT4 ratio provides a relative measure of translocation efficiency. At baseline (0 h), the observed ratio ranged between 0.05 and 0.08 (5–8%), indicating a typical basal surface GLUT4 level. The peak translocation

ratio reached approximately 0.15–0.20 (15–20%), suggesting that about 20% of the total GLUT4 pool translocated to the muscle cell surface upon insulin stimulation. The predicted vs actual GLUT4 ratio plot (Fig. S3) confirms that the linear mixed-effects model accurately predicts GLUT4 translocation, accounting for biological variation among triplicates. Furthermore, the residual diagnostics plot (Fig. S3) indicates appropriate model fit and reliability. Thus, this statistical model effectively demonstrates how total GLUT4 content influences the proportion of GLUT4 present at the cell membrane over time (Fig. S3), highlighting physiologically relevant GLUT4 dynamics.

### *Per3* knockdown impairs GLUT4 translocation in L6 cells following siRNA-mediated gene silencing

To investigate the role of regulatory genes in GLUT4 translocation, siRNA-mediated knockdown was performed targeting *Per3*, *Tssk6*, *Arntl*, and *Hoxb5* in L6 cells. The efficiency of gene knockdown was validated using RT-qPCR (Fig. 4A). Significant reductions in

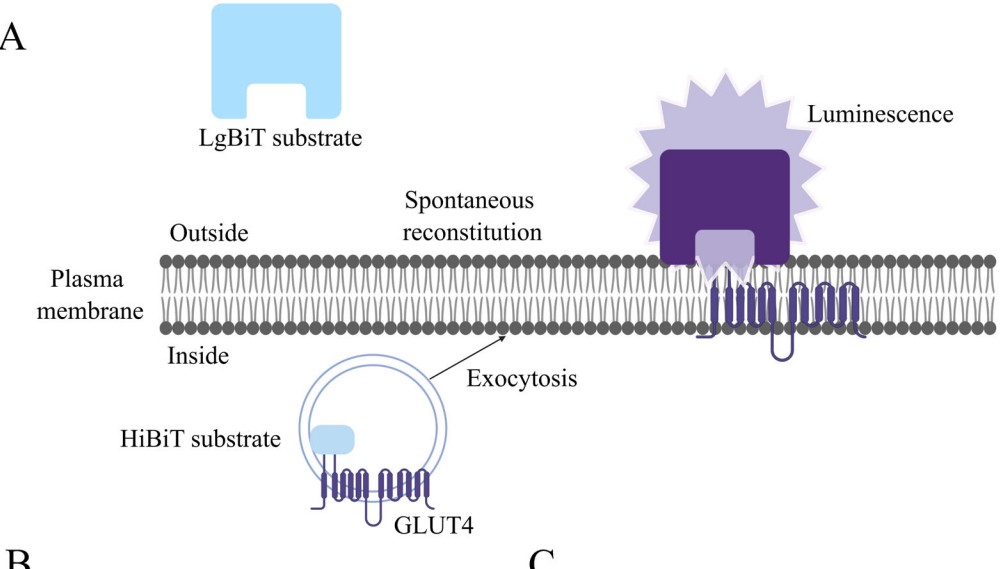

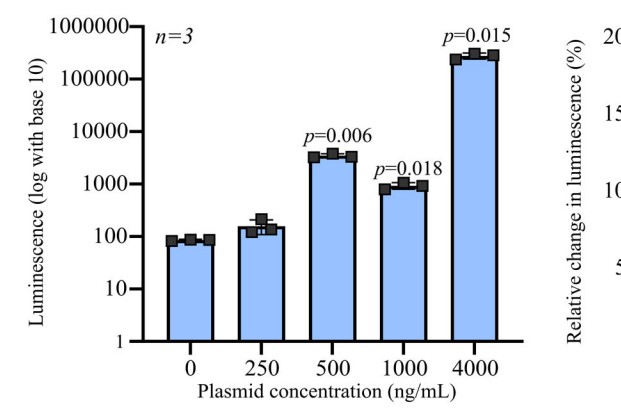

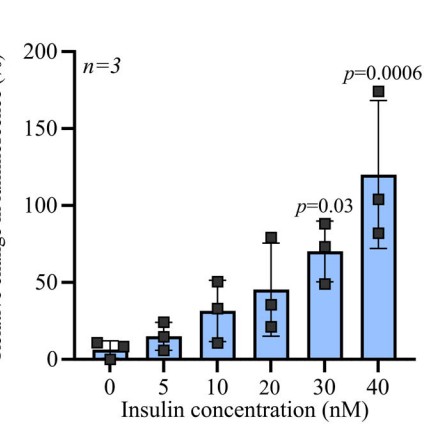

**Fig. 2. Luminescence detection of GLUT4-HibiT exocytosis upon insulin stimulation.** (A) Schematic representation of luminescence detection of surface GLUT4 upon exocytosis using spontaneous reconstitution of HiBiT and LgBiT substrate. (B) Luminescence detection of entire GLUT4 in cytoplasm after lysis of the cells, across various concentrations of DNA used for transfection. Luminescence showing $10^4$-fold increase normalised to non-transfected cells. Data are presented as mean±s.d. (*n*=3 biological replicates). (C) Luminescence measurement upon various concentrations of insulin. Data are presented as mean±s.d. (*n*=3 biological replicates). Data analysed by one-way ANOVA with Dunnett's post hoc test.

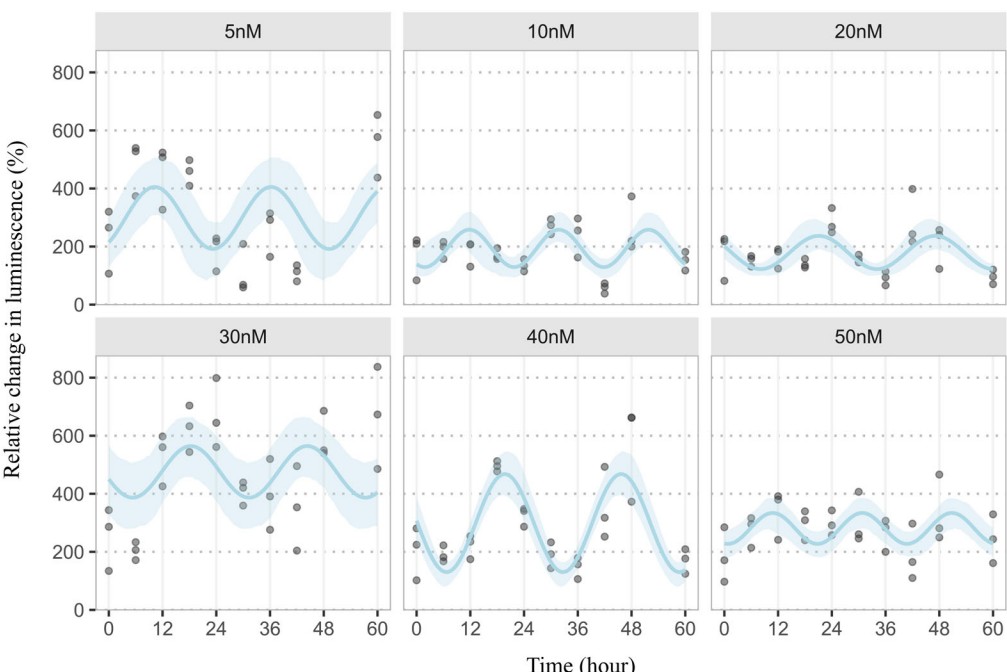

**Fig. 3. Circadian rhythm of GLUT4 translocation.** The circadian patterns of GLUT4 translocation in L6 skeletal muscle cells across varying insulin concentrations, analysed over 60 h. Data plotted using R (v4.2.1) with a harmonic regression script. Statistics are presented in Table 2 with *n*=3.

relative mRNA expression were observed for *Per3*, *Tssk6*, and *Arntl* (*P*<0.01) in response to silencing. Next, the effect of gene silencing on GLUT4 translocation was assessed under insulin stimulation (30 nM). As shown in Fig. 4B, knockdown of *Per3* resulted in a significant reduction in GLUT4 translocation, measured as luminescence relative to control cells (*P*<0.001). Similar reductions were observed following *Arntl* knockdown (*P*<0.01), whereas *Tssk6* knockdown showed no significant effect (*P*>0.05). These results highlight *Per3* and *Arntl* as novel regulators of GLUT4 trafficking in skeletal muscle cells.

### *Per3* knockdown disrupts GLUT4 translocation in a circadian rhythm-dependent manner in skeletal muscle cells

Given that *Per3* displayed the most robust reduction of GLUT4 translocation, we selected this as a target to study in a circadian time-course assay. Time-course analysis revealed that GLUT4 translocation was disrupted in *Per3* knockdown cells compared to scrambled control cells under 30 nM insulin stimulation (Fig. 5B). The rhythmic pattern observed in the control group was abolished in *Per3*-deficient cells, suggesting that *Per3* is essential for maintaining the circadian rhythm-dependent regulation of GLUT4 trafficking (Fig. 5B). To explore the metabolic consequences of *Per3* knockdown, glucose uptake assays were conducted. In human cells, the knockdown of *PER3* resulted in significantly reduced glucose uptake under insulin-stimulated conditions, compared to

non-targeting controls (Fig. 5C, *P*<0.01). Collectively, these results highlight the critical role of *Per3*/*PER3* in coordinating circadian rhythm-regulated GLUT4 translocation and metabolic functions, with significant implications for understanding glucose uptake mechanisms in both murine and human systems.

### Network inference reveals conditional dependencies between circadian genes and metabolic regulators

The circadian gene expression network estimation produced a network with 18 edges (27% saturation), revealing a tightly connected network of direct gene–gene associations. Bootstrap analysis confirmed that all edges were also statistically significant (*P*-adj<0.05), including the robust PER3–GLUT4 connection [*P*-adj=0.002, 95% CI (−0.46, −0.26)]. Across partial–correlation thresholds $|\rho| \in [0.1, 0.3]$, the core network structure remained qualitatively unchanged, underscoring the stability of the principal connection motifs under network stress. Additional partial correlations were observed among core clock genes including *PER2*, *CLOCK*, *BMAL1*, and *HOXB5*, highlighting coordinated expression patterns consistent with circadian regulatory architecture (Fig. 6).

### DISCUSSION

We have identified four promising targets that have intrinsically different cycling patterns between skeletal muscle cells from people with T2D and those who are matched for BMI and age but have NGT. These targets were also all associated with T2D in PheWAS studies (Scott et al., 2017; Suzuki et al., 2019; Xue et al., 2018). To further test the relevance of these targets in the pathology of T2D, we have developed a high-sensitivity, high-throughput, live-cell, circadian assay to measure GLUT4 translocation of skeletal muscle cells. With this new tool, we can perform high-throughput circadian assays of novel regulators of circadian insulin sensitivity in skeletal muscle. Our assay revealed *Per3* as a regulator of GLUT4 translocation in response to insulin, regulating glucose uptake, and circadian insulin sensitivity in skeletal muscle cells.

Our study illuminates the relationship between disrupted circadian rhythms and T2D, particularly in the context of skeletal

**Table 2. Overview of the JTK analysis for different insulin concentrations**

| Concentration (nM) | BH. Q | ADJ.P | PER | LAG | AMP |
|---|---|---|---|---|---|
| 5 | 0.002 | **0.002** | 24 | 17 | 1.42×10⁻¹⁴ |
| 10 | 1 | 1 | 24 | 12 | 2.49×10⁻¹⁴ |
| 20 | 1 | 1 | 24 | 6 | 1.78×10⁻¹⁴ |
| 30 | 0.003 | **0.003** | 24 | 9.5 | 184.57 |
| 40 | 0.53 | 0.53 | 24 | 8.5 | 1.07×10⁻¹⁴ |
| 50 | 0.3 | 0.3 | 24 | NA | NA |

NA, not applicable. Bold indicates significance.

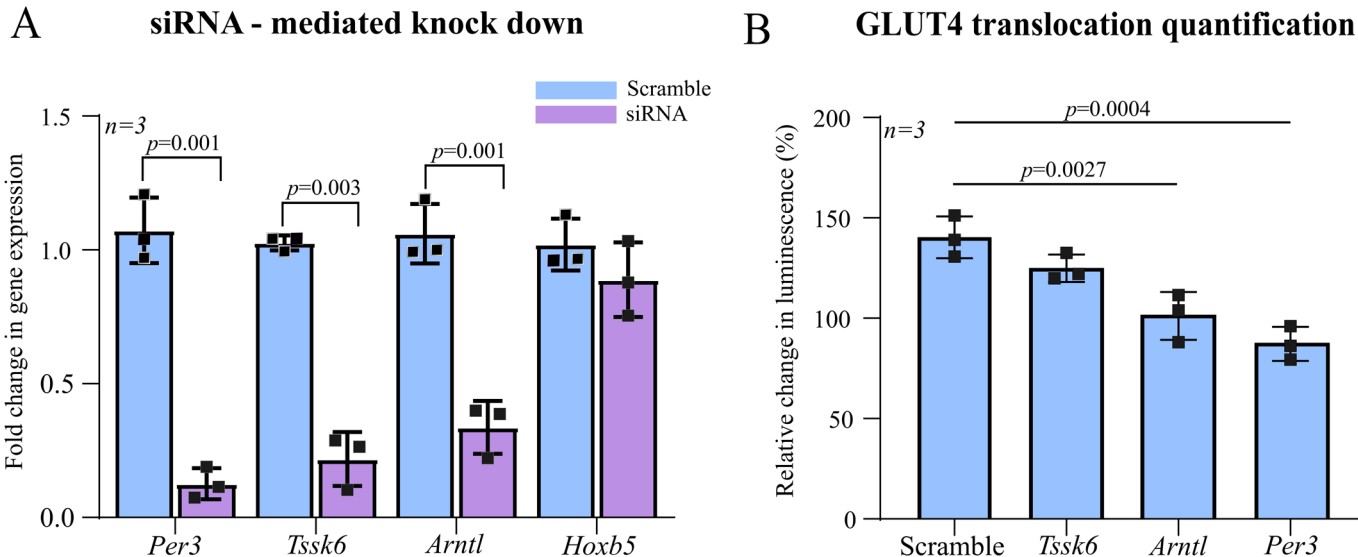

**Fig. 4. Effect of target gene knockdown on GLUT4 translocation.** (A) Validation of siRNA-mediated knockdown efficiency for *Per3*, *Tssk6*, *Arntl*, and *Hoxb5* genes in L6 cells. Relative gene expression was quantified using qPCR and normalised to control (untreated cells). Data are presented as mean±s.d. (*n*=3 biological replicates). Unpaired *t*-tests with Welch's correction were performed for each gene (*n*=3 per group). A significant knockdown was observed for *Per3*, *Tssk6*, and *Arntl* compared to control (*P*<0.01). (B) Quantification of GLUT4 translocation in response to 30 nM insulin stimulation following siRNA knockdown of *Per3*, *Tssk6*, and *Arntl*. Translocation was measured as the relative change in luminescence compared to control cells. Ordinary one-way ANOVA with Dunnett's post hoc test comparing each siRNA-treated group to Scramble control (*n*=3/group). *P*<0.05 was considered significant. Knockdown of *Per3* and *Arntl* significantly reduced GLUT4 translocation, while *Tssk6* knockdown caused a moderate but statistically non-significant reduction (*P*>0.05). Data are presented as mean±s.d. (*n*=3 biological replicates).

muscle dysfunction. Skeletal muscle's pivotal role in glucose homeostasis and insulin resistance in T2D is well established, underscored by persistent defects in insulin-stimulated glucose uptake and mitochondrial oxidation in myocytes obtained from individuals with T2D (Gabriel et al., 2021). A significant aspect of our findings lies in the characterisation of intrinsic differences in cycling patterns between cells derived from individuals with T2D and age- and BMI-matched controls exhibiting NGT. It has previously been demonstrated that there is a relationship between circadian clocks and insulin resistance, a key component of T2D (Stenvers et al., 2019).

Previous work has shown that insulin sensitivity is impaired in myoblasts derived from individuals with T2D, including induced pluripotent stem cell-derived models, which exhibit reduced glucose uptake in response to insulin (Batista et al., 2020). While circadian disruption has been implicated in T2D pathophysiology, the specific genes involved in regulating insulin sensitivity within skeletal muscle remain poorly defined. This gap is partly due to the lack of dynamic tools for assessing circadian regulation of GLUT4 translocation.

To address this, we developed a high-sensitivity, live-cell assay capable of detecting temporal changes in GLUT4 translocation in

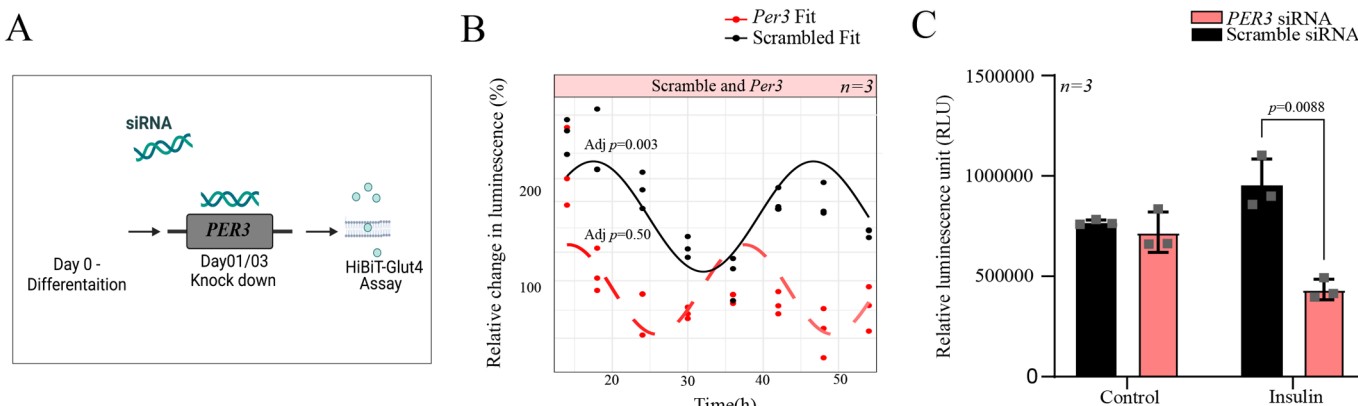

**Fig. 5. Role of *Per3* in regulating GLUT4 translocation and glucose uptake.** (A) Systematic representation of siRNA-mediated knockdown in L6 cells using Lipofectamine RNAiMAX. (B) Time-course analysis of GLUT4 translocation under 30 nM insulin stimulation in *Per3* knockdown and scrambled control cells. Data represent the percentage change in luminescence over time, with fitted curves illustrating the distinct patterns of GLUT4 response between *Per3* knockdown (red) and control (black). Statistical significance was determined using a JTK analysis with *Per3* knockdown losing the rhythm. (C) Glucose uptake in human cells with *PER3* knockdown compared to non-targeting control cells. Cells were treated with or without 30 nM insulin, and glucose uptake was measured as relative luminescence units (RLU). Knockdown of *PER3* significantly reduced glucose uptake under insulin-stimulated conditions. Data are presented as mean±s.d. (*n*=3 biological replicates). Statistical analysis was performed using two-way ANOVA. Statistics were performed using GraphPad Prism (10.4.1).

## Circadian Gene Network in Human Skeletal Muscle

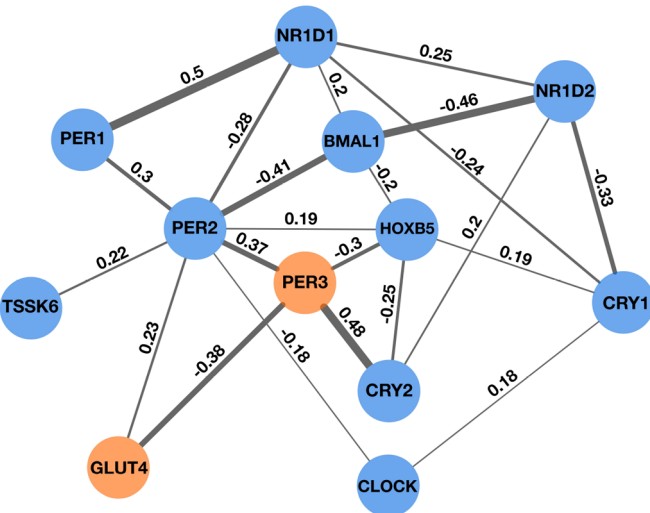

**Fig. 6. Gaussian graphical model of circadian gene expression in human skeletal muscle in T2D and control samples (data from Gabriel et al., 2021).** Nodes represent genes and edges indicate regularised partial corrections between expression logCPM, conditioned on circadian genes presented. Network model selection minimised the EBIC gamma=0.3 to balance complexity and sparsity. Edges with absolute partial correlations greater than 0.18 optimised model fit and were retained with edge width proportional to absolute partial correlation. Blue nodes denote core circadian genes; orange node highlights PER3 and GLUT4.

response to relatively low insulin concentrations. Unlike traditional endpoint assays requiring cell fixation (Heckmann et al., 2022; Wang et al., 1998, 2020), our platform enables continuous, real-time assessment of translocation in the same population of cells. In the current study, insulin dose-response experiments revealed a linear increase in GLUT4 translocation with increasing insulin concentrations, with saturation at 40 nM and a maximal translocation response of 120%. This is a higher sensitivity in response to relatively low insulin exposure when compared to end-point assays (Heckmann et al., 2022; Wang et al., 1998, 2020). This enhanced sensitivity allowed us to detect dose-dependent effects and identify 30 nM insulin as an optimal concentration for probing circadian rhythmicity.

Building on this, we screened circadian genes previously identified as dysregulated in T2D skeletal muscle (Gabriel et al., 2021) and found that knockdown of *PER3* significantly reduced GLUT4 translocation and abolished its rhythmicity. These findings support a mechanistic role for *PER3* in circadian insulin sensitivity and highlight the utility of our assay for identifying functional circadian regulators in skeletal muscle.

These genes were significantly associated with T2D in PheWAS datasets, highlighting their potential role in circadian dysregulation linked to metabolic disease. Gene silencing experiments further established *Per3* and *Arntl* as regulators of GLUT4 translocation. Interestingly, knockdown of *Per3* resulted in the most pronounced reduction in GLUT4 translocation, and *Per3* was therefore selected as the target to take forward into a circadian GLUT4 translocation assay. However, it should be noted that knockdown efficiency of *Arntl* was lower than that of *Per3*, which may influence the magnitude of effect on GLUT4 translocation, and *Arntl*/Bmal1 has previously been implicated as a key modulator of glucose tolerance and insulin sensitivity (Dyar et al., 2014; Schiaffino et al., 2016).

Previous research has suggested no obvious circadian disruption in *PER3* rhythmic gene expression in myocytes from participants with T2D and healthy participants (Hansen et al., 2016), albeit with lower statistical power ($n$=3) than the data analysed in the current study ($n$=5-7) (Gabriel et al., 2021). However, in the current study, *Per3* ablation impaired circadian rhythmicity, emphasising its essential role in maintaining circadian-regulated insulin sensitivity. This circadian disruption was at a much greater magnitude (Fig. 5A) when compared to the circadian gene expression disruption of *Per3* between T2D and NGT (Fig. 1). Additionally, glucose uptake assays confirmed that *PER3* knockdown impaired glucose utilisation under insulin-stimulated conditions in human skeletal muscle cells.

While our study suggests a potential link between disrupted circadian rhythms and insulin resistance in skeletal muscle, further investigations are warranted. It is noteworthy that significant GLUT4 rhythmicity was observed only at 5 nM and 30 nM insulin, but not at intermediate concentrations. While the reasons for this non-linear dose response are not yet clear, it may reflect threshold effects in circadian regulation of insulin signalling. This pattern should be interpreted with caution and warrants further investigation in future studies. Furthermore, it is not clear how these data link to glycolytic and mitochondrial metabolism, and disrupted circadian metabolism plays a role in loss of appropriate temporal response in these pathways (Gabriel et al., 2021). Understanding the precise mechanisms underlying these intrinsic differences in cycling patterns and their implications for insulin sensitivity, particularly within the context of T2D, remains a critical avenue for future research. To extend our mechanistic understanding, we conducted a new circadian gene network analysis that identified *PER3* and *ARNTL* as central nodes within the regulatory architecture controlling insulin sensitivity and GLUT4 trafficking. These genes showed strong partial correlations with components of vesicle transport and metabolic signalling pathways – key processes required for GLUT4 mobilisation. This systems-level insight reinforces their functional relevance and supports a model in which PER3 and ARNTL integrate circadian and metabolic cues to optimise insulin-stimulated glucose uptake. However, the precise mechanisms by which circadian regulators such as PER3 and ARNTL inhibit or facilitate GLUT4 translocation remain unclear. It is plausible that mitochondrial crosstalk with the core clock plays a role (Gabriel and Zierath, 2022). These findings also suggest that circadian disruption may impair insulin-stimulated GLUT4 translocation through downstream effects on both mitochondrial and glycolytic metabolism, which are key regulators of vesicle trafficking and fusion. GLUT4 translocation is a highly energy-dependent process requiring coordinated cytoskeletal rearrangement, vesicle tethering, and membrane fusion – all of which rely on adequate ATP supply, localised $Ca^{2+}$ signalling, and redox homeostasis (Bryant et al., 2002; Fazakerley et al., 2019). Mitochondria, under circadian control in skeletal muscle (Gabriel et al., 2021; Peek et al., 2013), contribute not only to ATP production but also play a critical role in regulating $Ca^{2+}$ signalling in skeletal muscle (Gabriel and Zierath, 2022; Ivarsson et al., 2019), and $Ca^{2+}$ promotes GLUT4 exocytosis and reduces its endocytosis (Li et al., 2014). Given the direct regulation of mitochondrial metabolism by the core clock in skeletal muscle (Lassiter et al., 2018), it is plausible that this regulatory loop is disrupted by *Per3* ablation. Conversely, family member *Per2* is regulated by skeletal muscle contraction in a calcium-dependent manner (Small et al., 2020), suggesting a possible bidirectional feedback loop. Additionally, glycolytic flux has been shown to influence actin remodelling and GLUT4 vesicle movement along the cytoskeleton (Stöckli et al., 2011). Circadian regulation of glycolytic

enzymes may therefore determine temporal windows of enhanced GLUT4 translocation efficiency (Dyar et al., 2014; Huang et al., 2011). In this context, PER3 may modulate insulin sensitivity by coordinating the timing of mitochondrial and glycolytic activity to optimise GLUT4 trafficking. The loss of rhythmic GLUT4 translocation following PER3 knockdown supports this hypothesis and suggests that misalignment between the circadian clock and cellular metabolism compromises insulin action at the level of vesicle mobilisation and membrane insertion. Overall, translating these insights into viable therapeutic strategies necessitates a comprehensive grasp of the regulatory networks governing circadian fluctuations in skeletal muscle insulin sensitivity.

Thus, our findings suggest that targeting the skeletal muscle clock may represent a novel strategy to restore insulin sensitivity in individuals with metabolic disease. Recent studies have shown that exercise training can influence the molecular clock in skeletal muscle, even in metabolically impaired states. For example, (Harmsen et al., 2024) demonstrated that training alters the expression and amplitude of key clock genes such as *Per1*, *Per2*, and *Rorα*, supporting a role for exercise as a peripheral circadian entrainer. This study also found that while exercise training modulates skeletal muscle clock gene expression in men with insulin resistance, it does not restore 24-h rhythmicity in substrate metabolism. These findings suggest that clock gene responsiveness to behavioural cues like exercise may precede full restoration of circadian metabolic function and highlight the importance of timing in intervention strategies targeting insulin sensitivity. This is consistent with recent translational work from our group demonstrating that morning exercise in combination with pre-breakfast metformin significantly improves glycaemic control in individuals with T2D (Carrillo et al., 2024), and further supported by our proposed framework for circadian-aligned timing of exercise, nutrition, and pharmacotherapy to optimise metabolic health (Peña Carrillo et al., 2024). Importantly, our new circadian gene network analysis positions PER3 and ARNTL as central regulators of insulin sensitivity in skeletal muscle, providing systems-level support for their functional role in GLUT4 translocation.

In summary, we have identified disrupted circadian rhythms in key genes (*PER3*, *ARNTL*, *TSSK6*, *HOXB5*) in skeletal muscle cells from individuals with T2D, advancing current understanding of how intrinsic timekeeping mechanisms contribute to insulin resistance. Through the development and application of a novel, high-sensitivity live-cell GLUT4-HiBiT assay, we demonstrate that PER3 is essential for maintaining circadian-regulated GLUT4 translocation and insulin-stimulated glucose uptake. PER3 knockdown abolished GLUT4 rhythmicity and impaired insulin sensitivity, establishing a direct functional link between circadian dysregulation and glucose metabolism. These findings offer a mechanistic basis for circadian disruption in T2D and underscore the translational potential of targeting the skeletal muscle clock to improve metabolic health. This work provides both conceptual and methodological advances that may inform future therapeutic strategies and public health interventions aimed at restoring insulin sensitivity through circadian alignment.

## MATERIALS AND METHODS
### Identifying novel regulators for circadian insulin sensitivity in skeletal muscle
In this study, initial data preprocessing involved the application of the 'filterByExpr' function from the 'edgeR' package, which was utilised to selectively exclude features with low expression levels, thereby enhancing the robustness of the dataset. Subsequently, the 'removeBatchEffect'

function from the 'limma' package was employed to correct for any potential batch effects, ensuring that observed variations were attributable to biological differences rather than technical artifacts. Post-preprocessing, the data were converted to log counts per million (logCPM) values for each detected feature, facilitating a standardised comparison across samples. This processed dataset, including all features, is publicly accessible via the NCBI repository under the Gene Expression Omnibus (GEO) accession code GSE182117. For analytical purposes, the data were extracted and normalised using the base R scale function. This normalisation process transformed the data into Z-scores, thereby standardising them for more effective comparative analyses and visualisation. PheWAS analysis was conducted using the ATLAS PheWAS database. Genetic variants associated with diabetes-related traits were queried, and statistical associations with multiple phenotypes were retrieved. RNA-sequencing data from synchronised primary myotube cultures (GSE182117) were adapter- and quality-trimmed using Trim Galore (v0.6.10), then aligned to GRCh38.p14 using STAR v2.7.11a in two-pass mode with GENCODE v44 annotation, achieving >87% uniquely mapped reads. Gene-level counts were generated using featureCounts (Subread v2.1.0) for unstranded data with negligible differences in gene-level counts (<1%), consistent with the original dataset (Gabriel et al., 2021). Counts were filtered using filterByExpr (edgeR v4.4.2), normalised to log2CPM, and batch-corrected via removeBatchEffect (limma v3.62.2). Rhythmicity was assessed using JTK_Cycle (MetaCycle v1.2.0).

### Cell culture conditions and compounds
L6-GLUT4 HiBiT cells were grown in MEMα supplemented with 10% fetal bovine serum (FBS) (PAA, Tet free) and 2.5 µg/ml blasticidin (Invitrogen). L6-GLUT4 HiBiT cells were incubated in starvation media 24 h prior to each experiment and serum shocked for circadian experiments. For differentiation, the cells were grown in MEMα supplemented with 2% horse serum (Lonza) for 7 days. Human skeletal muscle myoblasts were grown in Gibco™ Dulbecco's modified Eagle medium (DMEM)/F-12, HEPES supplemented with human epidermal growth factor (hEGF) (3 µg/ml), dexamethasone (0.1%) and FBS (10%). For differentiation, the cells were grown in Gibco™ DMEM/F-12, HEPES supplemented with 2% horse serum (Lonza) for 7 days. All cells were grown at 37°C and 5% $CO_2$.

### Cell line creation – transfection of GLUT4-HiBiT
In this study, we developed a stable myocyte cell line that expresses GLUT4-HiBiT in skeletal muscle cells. Our choice of L6 rat skeletal muscle cells (ATCC, CRL-1458) was guided by their high insulin sensitivity, making them particularly suitable for insulin sensitivity assays (Abdelmoez et al., 2020). The expression vector, TK-HiBiT-GLUT4 (Fig. S1), was provided by Promega UK. We followed the transfection procedure outlined below (Prosen et al., 2013). For the selection of stable clones, we applied 2.5 µg/ml blasticidin (Invitrogen).

### Magnetofection
For each transfection, 4 µg DNA was diluted in 200 µl serum-free media. Prior to each use, the PolyMag or PolyMag Neo tube was vigorously vortexed. 15 µl of PolyMag was added to a microtube or a U-bottom microwell. The 200 µl of DNA solution was then immediately mixed with the PolyMag or PolyMag Neo solution through vigorous pipetting. The mixture was incubated for 20 min, allowing for the formation of the transfection complex. The transfection mix (DNA+PolyMag or PolyMag Neo) was then added to the cells in T25 flask. The cell culture plate (T25) was placed upon a magnetic plate for a duration of 20 min to facilitate transfection.

### Stable cell line generation (post transfection)
Cells were incubated at 37°C in a $CO_2$ incubator for 48 h to allow for the expression of the HiBiT vector. After 48 h, the medium was replaced with fresh culture medium containing 1.0 µg/ml (lower-concentration) blasticidin (Invitrogen). The medium containing blasticidin was replaced every 2-3 days (Fig. S2). Cells were monitored daily, and the selection process was continued with an increase in the concentration of blasticidin until colonies of blasticidin-resistant cells were visible, typically within 2-3 weeks. Stable clones were screened for the expression of HiBiT using

Nano Glow HiBiT Assay. One-way ANOVA followed by Dunnett's post hoc test (GraphPad Prism 10.4.1) was applied for statistical analysis, considering $P<0.05$ as statistically significant.

### Insulin dose response in myotubes using Nano-Glow HiBiT assay

L6-GLUT4-HiBiT cells were resuspended to $5\times1000$ cells in 100 µl growth medium, plated in 96-well Greiner flat transparent plates. After myotube formation, cells were serum starved for 24 h. On the day of the experiment, the cells were washed twice with PBS, and 100 µl serum-free MEMα was added with 5 mM glucose (Sigma-Aldrich, filter sterilised). The cells were treated with gradient dose of insulin (0, 5, 10, 20, 30, 40) (stock: 1 mg/ml) and immediately screened. Translocation of GLUT4 from the intracellular storage pool to the plasma membrane was quantified using Nano-Glow HiBiT detection tool from Promega UK. An equal volume of Nano-Glo HiBiT Reagent (100 µl) (Promega, N3030), consisting of Nano-Glo HiBiT Buffer, Nano-Glo HiBiT Substrate, and LgBiT Protein, was added according to the manufacturer's protocol, and cells were incubated for 15 min at room temperature with shaking. Luminescence was measured using a Tecan Spark® 10 M plate reader (Tecan, Männedorf) controlled by SparkControl™ V2.0 software with 0.5 s of integration time at room temperature. Data were normalised to the value of untransfected cells (no GLUT4-HiBiT). One-way ANOVA followed by Dunnett's post hoc test (GraphPad Prism 10.4.1) was applied for statistical analysis, considering $P<0.05$ as statistically significant.

### Real-time luminescence recording in myotubes – 60 h

After postfusion, myotubes were synchronised with serum shock (50% FBS+MEMα, supplemented with 2.5 µg/ml Blasticidin) for 2 h in 37°C in a cell culture incubator, then the medium was changed to 100 µl serum-free MEMα with 5 mM glucose (Sigma-Aldrich, filter sterilised) to create the low-glucose condition. Luminescence was measured every 6 h for over 60 h period using Nano-Glo-HiBiT detection assay. Fresh medium was replenished at the end of every reading. Luminescence was normalised to the value of untransfected cells (no GLUT4-HiBiT). Error bars, s.d. ($n=3$).

### Rhythmicity analysis

Rhythmicity was assessed using JTK cycle using JTK_CYCLE in R. The analysis was conducted using the 'JTK_CYCLEv3.1' R script. The R (v4.2.1) environment was set to handle strings as non-factors. Two data files, 'x.annot.txt' and 'x.txt', were imported. The search for circadian rhythms was narrowed to periods of 60 h, corresponding to five time points per cycle. This was set using the 'periods' variable. The results were converted to a data frame and then subjected to Benjamini-Hochberg (BH) correction for multiple testing using the 'P.adjust' function. The corrected P-values (BH. Q), alongside the JTK_CYCLE output (ADJ.P, PER, LAG, AMP), were collated with the annotation data. The results were sorted based on adjusted P-value and amplitude. The scripts used are publicly available at https://github.com/RashmiSivasengh/circadian-rhythm.

### Efficiency of GLUT4 translocation – mixed effect modelling to access robust statistical inference of time-dependent translocation

Surface GLUT4 was calculated using HiBiT/LgBiT assay, and total GLUT4 was quantified using HiBiT/LgBiT assay after lysis of the cells using the reagent from Promega. To normalise for difference in overall GLUT4 expression among replicates or experimental conditions, a surface to total ratio was calculated: GLUT4 Ratio=Surface (GLUT4−Luminescence)/ Total GLUT4 (post lysis measurement). This ratio provides a relative measure of translocation efficiency (i.e. the fraction of total GLUT4 present on the cell surface). All statistical models were performed in R. A linear mixed-effect model was fitted using lm34 package to predict the GLUT4 surface to total ratio over time. The model includes timepoint and total GLUT4 as fixed effect, with sample as a random intercept to count for repeated measures and biological variability across surface: GLUT4_Ratio~Timepoint+Glutr.Total+(1|sample).

Model assumptions (linearity, normality, homoscedasticity of residuals) were assessed by visual inspection of diagnostic plots, including residuals vs fitted and quantile-quantile (Q-Q) plots. Pearson residuals were used to

evaluate how well the model accounted for variance at different levels of predicted values. Model performance was further assessed by comparing predicted vs observed GLUT4 ratios and computing correlation coefficients (e.g. Pearson's $r$).

### siRNA-mediated knockdown and real-time polymerised chain reaction

Commercially available siRNA targeting PER3 (human cells)/Per3, Arntl, Tssk6 and Hoxb5 were purchased from Thermo Fisher Scientific. L6 skeletal muscle myocytes were cultured with MEMα and 10% FBS. Cells were maintained at 37°C in a humidified atmosphere with 5% $CO_2$. Cells were plated at 60–80% confluence using trypsin-EDTA (Gibco, Thermo Fisher Scientific). Cells were differentiated at 90% confluency and transfected at day 1 of differentiation. L6 cells were transfected using Lipofectamine RNAiMAX (Invitrogen, Thermo Fisher Scientific) following the manufacturer's protocol (Invitrogen, 2020). A stock solution of siRNA was prepared at a concentration of 40 µM. For each well of a six-well plate, 60 µl OPTI-MEM (Gibco, Thermo Fisher Scientific) was mixed with 3.6 µl Lipofectamine RNAiMAX. In a separate tube, 50 µl OPTI-MEM was combined with 5 µl siRNA, resulting in a final siRNA concentration of 200 pmol/l. After gentle mixing, 55 µl from the first tube containing the Lipofectamine RNAiMAX mixture was added to the siRNA solution. The combined mixture was then incubated at room temperature for 15 min to allow the formation of siRNA-lipid complexes. Meanwhile, the cells were washed with phosphate-buffered saline (PBS), and 1 ml MEMα supplemented with 2% horse serum was added to each well. Subsequently, 100 µl of the siRNA final mixture was directly added to the cells, which were then incubated for 12 h. Following incubation, the medium was replaced with fresh differentiation medium. The cells were allowed to stabilise for 24 h before conducting further assays. Quantitative PCR (qPCR) was carried out for determining gene expression by using Fast SYBR Green Master Mix (Thermo Fisher Scientific) and predesigned TaqMan Gene Expression Assays (Thermo Fisher Scientific). The TaqMan rat housekeeping gene was $B2m$. Data are expressed as mean±s.d. from three independent biological replicates. For gene expression studies, qPCR data were normalised to untreated controls. Statistical differences between groups were determined using unpaired two-tailed $t$-tests with Welch's correction, and adjustments for multiple comparisons were made using the false discovery rate (FDR) method (Benjamini, Krieger and Yekutieli) with significance set at $P<0.01$. GLUT4 assay was carried out in the knockout cells with scramble as control. For GLUT4 translocation in response to 30 nM insulin, one-way ANOVA followed by Dunnett's post hoc test (GraphPad Prism 10.4.1) was applied for statistical analysis, considering $P<0.05$ as statistically significant.

### Glucose uptake in human skeletal muscle myotubes

The assay was performed using the glucose Uptake-Glo TM Assay kit (Promega), which provides a luminescent readout based on the detection of 2-deoxy glucose-6 phosphates (2DG6P), a glucose analogue, as an indicator of glucose uptake activity. Human cells were plated in a 96-well plate at a density of 10,000 cells per well and allowed to adhere overnight. Post fusion, *PER3* gene was silenced, the cells were serum-starved for 24 h, followed by glucose uptake assays. Statistical analysis was performed using two-way ANOVA. All statistics were performed using GraphPad Prism (10.4.1).

### Circadian gene network analysis

A curated subset of circadian genes was selected from the normalised expression matrix and annotated using the AnnotationDbi package (v1.68.0). Expression values were scaled and centred using base R (v4.3.1). The sample covariance matrix $\boldsymbol{\Sigma}$ was computed and used to estimate the precision matrix $\boldsymbol{\Theta}=\boldsymbol{\Sigma}^{-1}$ via the graphical lasso algorithm, implemented in the glasso package (v1.1.2). Regularisation was applied to promote sparsity in the estimated network of conditional dependencies. Model selection was guided by minimisation of the extended Bayesian information criterion (EBIC), balancing fit with model complexity. Partial correlations were derived from the precision matrix as $\rho_{ij\cdot\text{rest}}=-\frac{\theta_{ij}}{\sqrt{\theta_{ii}\,\theta_{jj}}}$,

Biology Open

ρij·rest=−θiiθjjθij, quantifying the direct association between gene pairs 'i' and 'j', conditioned on all other genes in the model. The resulting Gaussian Graphical Model was visualised using the igraph package (v2.0.3), applying a Fruchterman–Reingold force-directed layout with genes as nodes and regularised partial correlations as weighted edges.

## Statistical analysis

Data are presented as mean±s.d. from at least three independent biological replicates, and analyses were performed using GraphPad Prism 10.4.1 or R Studio (2024.12.1+563). Group comparisons for colony selection, insulin dose response, and GLUT4 translocation were conducted using one-way ANOVA followed by Dunnett's post hoc test ($P<0.05$). Gene expression data (qPCR) were normalised to untreated controls and analysed using unpaired two-tailed $t$-tests with Welch's correction. Glucose uptake assays were analysed by two-way ANOVA.

## Acknowledgements
We would like to thank Promega UK for their kind donation of the expression vector (TK- HiBiT-GLUT4), which we used to create a stable cell line in L6 cells.

## Competing interests
The authors declare no competing or financial interests. The expression vector was donated by Promega UK.

## Author contributions
Conceptualization: R.S., A.S., B.M.G.; Data curation: R.S., A.S., B.M.G.; Formal analysis: R.S., A.S., B.M.G.; Funding acquisition: B.M.G.; Investigation: R.S., A.S., B.M.G.; Methodology: R.S., B.M.G.; Project administration: B.M.G.; Resources: B.M.G.; Supervision: B.M.G.; Visualization: B.M.G.; Writing – original draft: R.S., B.M.G.; Writing – review & editing: R.S., A.S., B.M.G.

## Funding
B.M.G. was supported by a fellowship from Novo Nordisk Fonden (NNF19OC0055072). R.S. was supported by an Elphinstone Scholarship from the University of Aberdeen. Open Access funding provided by University of Aberdeen. Deposited in PMC for immediate release.

## Data and resource availability
All scripts used for analysis are publicly available at https://github.com/RashmiSivasengh/circadian-rhythm. Raw data are available at https://doi.org/10.6084/m9.figshare.28409729. All other relevant data can be found within the article and its supplementary information. GLUT4 HiBiT biological resources are available upon request.

## First Person
This article has an associated First Person interview with the first author of the paper.

## Peer review history
The peer review history is available online at https://journals.biologists.com/bio/lookup/doi/10.1242/bio.061941.reviewer-comments.pdf

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
