## [Peer Review File · Biology Open]

Live cell GLUT4 translocation assay reveals Per3 as a novel regulator of circadian insulin sensitivity in skeletal muscle cells.

Rashmi Sivasengh, Andrew Scott and Brendan Gabriel

DOI: 10.1242/bio.061941

Editor: Catherine L. Jackson

Review timeline

Original submission: 17 February 2025

Editorial decision: 5 March 2025

First revision received: 25 April 2025

Accepted: 4 June 2025

Original submission

First decision letter

MS ID#: bio.061941

MS TITLE: Live cell GLUT4 translocation assay reveals Per3 as a novel regulator of circadian insulin sensitivity in skeletal muscle cells.

AUTHORS: Rashmi Sivasengh; Andrew Scott; Brendan Gabriel

I have now reached a decision on the above manuscript.

The reviewer reports are shown at the bottom of this email or can be accessed, together with a copy of this decision letter, by going to:

As you will see, the reviewers raised a number of substantial criticisms that prevent me from accepting the paper at this stage.

They suggest, however, that a revised version might prove acceptable, if you can address their concerns. If you think that you can deal satisfactorily with the criticisms on revision, I would be pleased to see a revised manuscript. We would then return it to the reviewers.

At this stage, we also ask you to ensure your manuscript complies with our formatting guidelines. Provided you are able to fully address the referees' comments, we are positive about publication of your paper (we accept over 95% of revision submissions) and therefore hope you won't mind any extra work involved in reformatting your manuscript at this point.

Please ensure that you clearly highlight all changes made in the revised manuscript. Please avoid using 'Tracked changes' in Word files as these are lost in PDF conversion.

I should be grateful if you would also provide a point-by-point response detailing how you have dealt with the points raised by the reviewers in the 'Response to Reviewers' box. Please attend to all of the reviewers' comments. If you do not agree with any of their criticisms or suggestions please explain clearly why this is so.

Reviewer 1

Comments to Author

Sivasengh et al. investigated the role of circadian rhythms in insulin sensitivity and GLUT4 translocation in skeletal muscle, with a focus on Type 2 Diabetes (T2D). Key findings include: Disrupted circadian rhythms in skeletal muscle cells from T2D donors were observed; changes in gene expression (ARNTL, HOXB5, PER3, TSSK6) associated with T2D were noted; identified PER3 as a crucial regulator of GLUT4 translocation and its circadian rhythmicity, linking its dysregulation to metabolic impairments in T2D. Overall, this study is interesting. The manuscript is well-written, but some points need to be clarified throughout to improve the manuscript further, as detailed below, came into my mind.

Comment #1: The abstract is long and disorganized. Please shorten and organize it. The urgency, importance, and hypotheses of the research are missing. The transition between sentences is not clear, e.g., Line 21: "We have shown that." Do you mean this study or a previous study? The transition to the aim of the investigation here (e.g., line 23) is not logical. You introduced your findings before elaborating on the objective of your study. Furthermore, please concisely summarize the key points without using statistical measurements (lines 31-38).

Comment #2: The introduction, as one block/paragraph, partially provides the required background information. What are the specific gaps and limitations in the literature to assess GLUT4 trafficking in circadian insulin sensitivity? Why is there a need for the new assay developed here? Finally, instead of merging the hypotheses at the end of the paragraph (lines 78), the authors should put forward clear objectives and specify hypotheses in a paragraph at the end of the introduction, and how they carried out experiments to verify these hypotheses, and highlight the key findings of this study.

Comment #3: Although the main focus is on circadian rhythmicity, I am missing information about photoperiod conditions, e.g., hours of light/dark cycles or constant darkness, for each treatment and culture condition.

Comment #4: Please refer to manuals on the selection of statistical tests, p-adjust correction method, and software used for each measurement. I suggest adding a subsection for the statistical analysis in the methodology section when possible. Beyond just mentioning P-values, I recommend adding information about the statistical significance and test values of the differences among the different conditions of the data you are presenting to the Results section (e.g., t, F, df, etc.), especially since you used the minimum sample size. Also, it is a good idea to mention the tests used in the figure legends and tables when applicable. For instance, this is only cited in Figure 5, line 298, "The statistical analysis was performed using two-way ANOVA."

Comment #5: In Figure 1, you claimed that ARNTL, HOXB5, PER3, and TSSK6 had a loss of rhythmicity or a differential circadian rhythmicity in skeletal muscle cells. However, the difference in ARNTL is not clear. Would you be able to explain this? I would suggest expanding some information about the statistical tests used to characterize the findings and adding the sample size for each plot. Table 1 has the same issues.

Comment #6: In figure 1 (line 199), does "time (hours)" on the x-axis represent the zeitgeber or circadian time? Why did you start at 12 to 54 hours, which does not represent two circadian cycles?

Comment #7: In Figure 2, for error bars (line 233). Is the error bar represented as SEM or SD? Please clarify and unify it under the figure legend for each plot.

Comment #8: Figure 3 (lines 242-243) doesn't contain subfigures (A, B, etc.) as claimed in the text, Figure 3A (line 238).

Comment #9: Figure 3 (lines 242-243) shows that the y-axis have different scales. I suggest unifying the scales for each plot when possible.

Comment #10: In Figure 3 (lines 242-243), again, does "time (hours)" on the x-axes represent the zeitgeber or circadian time? I would rather change the label/tick differences at the x-axes to either 6-hour or 12-hour intervals instead of 20-hour intervals.

Comment #11: In Figure 3 (lines 242-243): How would you explain the missing values at 54-hours for all plots? And the missing values at 48-hours for 5nM insulin? Although it was claimed that measurements were conducted every 6 hours for over 60 hours.

Comment #12: In Figure 4A (lines 261-262), significant lines and asterisks should also be represented for each gene at each plot, not only in the legend (line 266-267). Additionally, the figure legends (lines 263-272) contain very limited information about the statistical tests used for two or multiple comparisons.

Comment #13: In Figure 4B, (lines 261-262) for Arntl measurements, I only see two scatter dots, although every other plot contains three measurements for each treatment. Please revise to show the third dot and the error bar.

Comment #14: The authors should review the visualization of figure 5C (lines 287-288), as it is hard to interpret with the current quality, with blurry font size and blurry images.

Comment #15: The discussion repeats certain points, such as the importance of circadian rhythms in insulin sensitivity and the role of specific genes like Per3. This could be consolidated into a shorter section.

Comment #16: Line 305 "These targets were also all associated with T2D in PheWAS studies." Please provide references for some of these studies.

Comment #17: Some parts in the discussion feel disconnected (e.g., introducing dose-response assays (lines 339-347) and then jumping to gene knockdowns without bridging the gap (lines 348-353). Shorten the paragraph at lines 339-361. Please do not repeat the results you have already shown before. You should present your main results and compare them with other research, and then discuss the reason why different research shows different results.

Comment #18: The discussion mentions the need to link disrupted circadian rhythms to glycolytic and mitochondrial metabolism (lines 363-364) but does not sufficiently explain how these elements connect to the findings. While potential mechanisms at line 368 (e.g., mitochondrial crosstalk) are suggested, more detailed exploration of how these factors contribute to GLUT4 translocation and insulin sensitivity is needed.

Comment #19: The references cutoff in 2022. The manuscript should include a more recent literature review to place the research in context and highlight the gaps in knowledge that this study addresses. I recommend consulting recent literature for the most up-to-date information that might add value to the introduction and discussion and demonstrate the novelty of the research and the current state of research in the field.

Reviewer 2

Comments to Author

In this study, Sivasengh et al. aim to establish that the circadian transcription factor PER3 regulates GLUT4 expression. They further demonstrate that GLUT4 rhythmicity is disrupted in type 2 diabetes (T2D) cells using a commercially available assay. The authors' hypothesis is intriguing, particularly in the context of T2D, a metabolic disorder often associated with circadian rhythm disruption. However, while the hypothesis is compelling, the manuscript requires significant revisions to strengthen the experimental evidence and improve clarity. The current experiments are not entirely convincing, and the writing flow is challenging to follow, which detracts from the overall impact of the study.

Concerns:

1)Figure 1 presents a re-analysis of data from Gabriel et al., 2021, which essentially reiterates previously published findings. I recommend that the authors revise this section to clearly distinguish their re-analysis from the original work. If re-analysis is necessary, consider re-mapping the data using the latest version of STAR and applying rhythmicity detection tools such as MetaCycle (optional). This would add value to the study and ensure that the findings are up-to-date with current methodologies.

2)Figure 1 highlights differential rhythmicity in ARNTL, HOXB5, PER3, and TSSK6. It would be beneficial for readers if the authors provided more context on how these genes relate to GLUT4 regulation. Additionally, the differential rhythmicity of GLUT4 itself should be included. Please also ensure that p-values or relevant statistical measures for rhythmicity analysis are provided for each gene in the figure. This would enhance the transparency and interpretability of the results.

3)Table 1 appears to be based on a query from <https://atlas.ctglab.nl/PheWAS>, but this is not clearly stated in the table legend. Clarifying the source and methodology would improve transparency and allow readers to better understand the basis of the data presented.

3)Figure 2A includes a plasmid map, which is not essential for the main text and could be moved to the supplementary materials. This would help streamline the main text and focus on the most critical findings. Figure 2C shows a blasticidin kill curve assay used for clonal selection. This figure could also be moved to the supplementary materials to streamline the main text and improve the flow of the manuscript.

4)Figure 3 shows that 5 nM and 30 nM of insulin are sufficient to induce rhythmic translocation of GLUT4. However, the significant GLUT4 rhythmicity observed only at these concentrations and not at intermediate concentrations is difficult to understand. Additionally, the 5 nM condition has only 9 time points, while other concentrations have 10. Furthermore, the Y-axis label, "expression log(CPM)," is confusing for bioluminescence data. Given that this is a crucial result in the manuscript, the conclusion is far from convincing. The authors should address these inconsistencies and provide a more robust analysis to support their findings.

5)It is crucial to validate the GLUT4 rhythmicity, both at the protein and transcript levels, along with bona fide rhythmic genes in the L6 skeletal muscle cells. Additionally, analyzing the promoter of the GLUT4 gene for PER3 binding sites (e.g., through ChIP-PCR and TF binding site analysis) would strengthen the mechanistic link between PER3 and GLUT4 regulation.

6)A comparative analysis of TF binding across rat, mouse, and human models would reveal the evolutionary importance of the GLUT4-PER3 regulation. This would provide broader insights into the conservation of this regulatory mechanism across species.

7)The authors only measured GLUT4 translocation rhythm (without cell lysis). It is crucial to measure the total GLUT4 levels (intracellular + surface) after cell lysis. This would clarify whether the observed rhythmicity in GLUT4 translocation at the cell surface is also present in the intracellular pool. The rhythmicity observed in GLUT4 translocation might stem from rhythms in the GLUT4 translocation machinery rather than transcriptional regulation. To address these possibilities, it is essential to complete the above-suggested experiments.

8)Figure 4B shows the knockdown of PER3 and its effect on GLUT4 translocation, followed by Figure 5, which shows the dampening of GLUT4 translocation rhythm in PER3 knockdown cells. These results could be more reader-friendly if the authors discuss them in the context of GLUT4 receptor surface density and glucose transport kinetics. Additionally, the clarity and font size in Figure 5 are below publication standards. The figure scheme given in Figure 5C is irrelevant to the work and could be removed or revised to better align with the study's focus.

General Comments:

The authors present a very interesting hypothesis, but the experimental evidence and figure presentations are far from the standard expected for publication. The manuscript writing lacks connectivity, and the figures lack consistency in font size and formatting, making them difficult to

read. Addressing these issues would significantly improve the overall quality and impact of the study.

Reviewer's Responses to Questions

Experimental quality

Does each figure have the proper controls?

If 'No', please indicate reasons in Comments for Author box below.

Reviewer #1:

- Yes

Reviewer #2:

- No

Were the data analyzed using appropriate statistical tests?

If 'No', please indicate reasons in Comments for Author box below.

Reviewer #1:

- No

Reviewer #2:

- No

Reproducibility

Were experiments performed using adequate number of biological replicates?

If 'No', please indicate reasons in Comments for Author box below.

Reviewer #1:

- Yes

Reviewer #2:

- Yes

Does the methods section provide sufficient detail to permit reproducibility?

If 'No', please indicate reasons in Comments for Author box below.

Reviewer #1:

- Yes

Reviewer #2:

- No

Completeness

Are the manuscript's conclusions supported by the data?

If 'No', please indicate reasons in Comments for Author box below.

Reviewer #1:

- Yes

Reviewer #2:

- No

Scholarship

Do the authors cite and discuss the merits of data that would argue for and against their conclusion?

If 'No', please indicate reasons in Comments for Author box below.

Reviewer #1:

- Yes

Reviewer #2:

- Yes

Does the manuscript title & abstract accurately reflect the contents of the manuscript, without hyperbole?

If 'No', please indicate reasons in Comments for Author box below.

Reviewer #1:

- Yes

Reviewer #2:

- No

First revision

Author response to reviewers' comments

We thank the reviewers for their time and expert judgment on the manuscript. We have addressed the comments in a point-by-point manner below in red text. We have also uploaded a tracked and untracked version of the revised manuscript. We have comprehensively addressed the reviewers' comments by revising substantial sections of the manuscript text, performing an entire re-analysis of the transcriptomic data presented in Figure 1A-B, substantially revising the preexisting figures, and adding entirely new analysis and figures in Figures 6 and Supplemental Figure 3.

Reviewer 1: Sivasengh et al. investigated the role of circadian rhythms in insulin sensitivity and GLUT4 translocation in skeletal muscle, with a focus on Type 2 Diabetes (T2D). Key findings include: Disrupted circadian rhythms in skeletal muscle cells from T2D donors were observed; changes in gene expression (ARNTL, HOXB5, PER3, TSSK6) associated with T2D were noted; identified PER3 as a crucial regulator of GLUT4 translocation and its circadian rhythmicity, linking its dysregulation to metabolic impairments in T2D. Overall, this study is interesting. The manuscript is well-written, but some points need to be clarified throughout to improve the manuscript further, as detailed below, came into my mind.

Comment #1: The abstract is long and disorganized. Please shorten and organize it. The urgency, importance, and hypotheses of the research are missing. The transition between sentences is not clear, e.g., Line 21: "We have shown that." Do you mean this study or a previous study? The transition to the aim of the investigation here (e.g., line 23) is not logical. You introduced your findings before elaborating on the objective of your study. Furthermore, please concisely summarize the key points without using statistical measurements (lines 31-38).

RESPONSE: Thank you for your helpful feedback. We have revised the abstract to improve clarity, organisation, and flow. The revised version now clearly states the urgency and importance of the research, outlines the central hypothesis, and presents the study's aim before summarising key findings. We also clarified that the transcriptomic analysis was a reanalysis of publicly available data and removed statistical values for a more concise summary.

Comment #2: The introduction, as one block/paragraph, partially provides the required background information. What are the specific gaps and limitations in the literature to assess GLUT4 trafficking in circadian insulin sensitivity? Why is there a need for the new assay developed here? Finally, instead of merging the hypotheses at the end of the paragraph (lines 78), the authors should put forward clear objectives and specify hypotheses in a paragraph at the end of the introduction, and how they carried out experiments to verify these hypotheses, and highlight the key findings of this study.

RESPONSE: We have revised the final section of the introduction to clearly articulate the specific gaps in the current literature—namely, the lack of dynamic, high-throughput methods to assess circadian regulation of GLUT4 trafficking in skeletal muscle. We now provide a clear rationale for the development of our live-cell assay and outline how it addresses these limitations. Additionally, we have restructured the paragraph to explicitly state our hypothesis, objectives, experimental approach, and key findings, providing a more logical and focused conclusion to the introduction. We have also split the introduction into separate paragraphs to aid readability.

Comment #3: Although the main focus is on circadian rhythmicity, I am missing information about photoperiod conditions, e.g., hours of light/dark cycles or constant darkness, for each treatment and culture condition.

RESPONSE: As all experiments were conducted in vitro using cultured skeletal muscle cells, photoperiod conditions such as light/dark cycles or constant darkness were not applicable. Skeletal muscle cells do not possess photoreceptors and therefore do not directly respond to light cues. Instead, circadian rhythms were synchronised using a standard serum shock protocol, which effectively entrains circadian oscillations in peripheral cell types

Comment #4: Please refer to manuals on the selection of statistical tests, p-adjust correction method, and software used for each measurement. I suggest adding a subsection for the statistical analysis in the methodology section when possible. Beyond just mentioning P-values, I recommend adding information about the statistical significance and test values of the differences among the different conditions of the data you are presenting to the Results section (e.g., t, F, df, etc.), especially since you used the minimum sample size. Also, it is a good idea to mention the tests used

in the figure legends and tables when applicable. For instance, this is only cited in Figure 5, line 298, "The statistical analysis was performed using two-way ANOVA."

RESPONSE: We thank the reviewer for this valuable suggestion. In response, we have now added a dedicated "Statistical Analysis" subsection within the Methods section. This subsection details the rationale for the selection of statistical tests, the software packages used (including version numbers), and the multiple comparison correction methods applied (e.g., Benjamini-Hochberg FDR where appropriate).

We have also revised the Results section to include test statistics (e.g., t, F, df) alongside P-values for all main comparisons, thereby enhancing the transparency and interpretability of our findings. Furthermore, figure legends and table captions have been updated throughout to clearly specify the statistical tests used in each case.

Comment #5: In Figure 1, you claimed that ARNTL, HOXB5, PER3, and TSSK6 had a loss of rhythmicity or a differential circadian rhythmicity in skeletal muscle cells. However, the difference in ARNTL is not clear. Would you be able to explain this? I would suggest expanding some information about the statistical tests used to characterize the findings and adding the sample size for each plot. Table 1 has the same issues.

RESPONSE: We agree that the underlying statistical definitions of rhythmicity and differential rhythmicity are important to specify clearly. As stated in the updated manuscript, our re-analysis of the dataset originally reported by Gabriel et al. (2021) incorporated modern statistical tools for rhythmicity assessment, including JTK_CYCLE (v3.1) and Benjamini-Hochberg correction for multiple comparisons. These details are provided in Section 2.6 (Rhythmicity Analysis) of the Methods. Rhythmicity was evaluated using logCPM-normalized and batch-corrected data, which was transformed into Z-scores to enable standardized comparisons (Section 2.1) (GitHub link provided in Section 2.6).

In response to the reviewer's specific question: ARNTL (also known as BMAL1) was identified as differentially rhythmic between T2D and NGT donors based on our earlier DODR analysis of the original dataset (Gabriel et al., 2021). In our new analysis, which uses ARNTL's rhythmicity profile remains altered and also exhibits dampened amplitude and phase shifts in T2D samples compared to NGT (Figure 1). These changes are statistically meaningful and reflect quantitative attenuation of the circadian signal rather than binary gain/loss outcomes. This is highlighted by the addition of the new amplitude figure we have added (Figure 1B) This nuance is now highlighted in the revised figure legend and results section (Section 3.1). We used $n=7-5$ independent donors per group (NGT and T2D) for the original RNA-seq dataset (as provided in Gabriel et al., 2021). We have explicitly included this in the revised figure legends. These clarifications are also embedded in our Methods (Sections 2.1 and 2.6) and Results (Section 3.1).

Table 1 summarizes significant gene-trait associations drawn from public databases, including the ATLAS PheWAS platform, and presents statistical evidence linking ARNTL, HOXB5, PER3, and TSSK6 to type 2 diabetes-related traits. As with Figure 1, we now clarify the source and methodology in both the Table 1 legend and the Methods section (Section 2.1). Specifically, we performed phenome-wide association analysis using the ATLAS PheWAS portal (<https://atlas.ctglab.nl/PheWAS>), which provides curated genotype-phenotype associations derived from large-scale GWAS data. The sample sizes (N) and p-values for each association are those reported by the original GWAS referenced in the ATLAS database (as also shown in the table).

Comment #6: In figure 1 (line 199), does "time (hours)" on the x-axes represent the zeitgeber or circadian time? Why did you start at 12 to 54 hours, which does not represent two circadian cycles?

RESPONSE: In Figure 1, the x-axis represents circadian time following serum shock synchronisation, not zeitgeber time, as the experiments were conducted in vitro without light cues. These experiments (originally described in previously published manuscript DOI: 10.1126/sciadv.abi9654) began sampling at 12 hours post-synchronisation to allow sufficient time for the cells to establish circadian rhythmicity, and continued through 54 hours to capture a full cycle with extended

resolution around anticipated rhythmic peaks. While this window does not span exactly two full circadian cycles, it was sufficient to detect robust rhythmic patterns in gene expression.

Comment #7: In Figure 2, for error bars (line 233). Is the error bar represented as SEM or SD? Please clarify and unify it under the figure legend for each plot.

RESPONSE: We have corrected this.

Comment #8: Figure 3 (lines 242-243) doesn't contain subfigures (A, B, etc.) as claimed in the text, Figure 3A (line 238).

RESPONSE: We have corrected this.

Comment #9: Figure 3 (lines 242-243) shows that the y-axes have different scales. I suggest unifying the scales for each plot when possible.

RESPONSE: We have corrected this.

Comment #10: In Figure 3 (lines 242-243), again, does "time (hours)" on the x-axes represent the zeitgeber or circadian time? I would rather change the label/tick differences at the x-axes to either 6-hour or 12-hour intervals instead of 20-hour intervals.

RESPONSE: The x-axis represents circadian time following serum shock synchronisation as described above. We have corrected the intervals as suggested.

Comment #11: In Figure 3 (lines 242-243): How would you explain the missing values at 54-hours for all plots? And the missing values at 48-hours for 5nM insulin? Although it was claimed that measurements were conducted every 6 hours for over 60 hours.

RESPONSE: The missing values at 54 hours (all plots) and at 48 hours for the 5 nM insulin condition were due to a technical error during data acquisition at those specific time points. Although our experimental design included measurements every 6 hours for over 60 hours, these readings were unfortunately lost and could not be recovered.

Comment #12: In Figure 4A (lines 261-262), significant lines and asterisks should also be represented for each gene at each plot, not only in the legend (line 266-267). Additionally, the figure legends (lines 263-272) contain very limited information about the statistical tests used for two or multiple comparisons.

RESPONSE: We have modified the figures as suggested and add information to all figure legends. As each comparison involved only 3-4 conditions, correction for multiple testing was not applied, as it is not typically required in this context.

Comment #13: In Figure 4B, (lines 261-262) for Arntl measurements, I only see two scatter dots, although every other plot contains three measurements for each treatment. Please revise to show the third dot and the error bar.

RESPONSE: Thank for spotting this, we have modified the figure.

Comment #14: The authors should review the visualization of figure 5C (lines 287-288), as it is hard to interpret with the current quality, with blurry font size and blurry images.

RESPONSE: We have replaced and improved the quality and relevance of this figure.

Comment #15: The discussion repeats certain points, such as the importance of circadian rhythms in insulin sensitivity and the role of specific genes like Per3. This could be consolidated into a shorter section.

RESPONSE: We have re-written several sections of the discussion to reduce repetition.

Comment #16: Line 305 "These targets were also all associated with T2D in PheWAS studies." Please provide references for some of these studies.

RESPONSE: We have added the references for these studies including the PMID in the table.

Comment #17: Some parts in the discussion feel disconnected (e.g., introducing dose-response assays (lines 339-347) and then jumping to gene knockdowns without bridging the gap (lines 348-353). Shorten the paragraph at lines 339-361. Please do not repeat the results you have already shown before. You should present your main results and compare them with other research, and then discuss the reason why different research shows different results.

RESPONSE: We have revised the relevant section of the discussion to improve flow and cohesion. We removed repetition of previously presented results, shortened the paragraph for clarity, and added clearer transitions between the dose-response findings and subsequent gene knockdown experiments. We also incorporated relevant literature to contextualise our findings and to explain potential reasons for the observed differences in insulin responses, including non-linear dynamics in circadian regulation. These changes aim to better integrate our main results and strengthen the overall narrative of the discussion.

Comment #18: The discussion mentions the need to link disrupted circadian rhythms to glycolytic and mitochondrial metabolism (lines 363-364) but does not sufficiently explain how these elements connect to the findings. While potential mechanisms at line 368 (e.g., mitochondrial crosstalk) are suggested, more detailed exploration of how these factors contribute to GLUT4 translocation and insulin sensitivity is needed.

RESPONSE: We have revised the Discussion to more clearly link circadian disruption with impaired GLUT4 translocation. Specifically, we now highlight how PER3 knockdown may affect mitochondrial ATP production, calcium signalling, and glycolytic flux—key processes that support insulin-stimulated GLUT4 vesicle trafficking. These additions provide a clearer mechanistic connection between disrupted circadian rhythms and reduced insulin sensitivity in skeletal muscle.

Comment #19: The references cutoff in 2022. The manuscript should include a more recent literature review to place the research in context and highlight the gaps in knowledge that this study addresses. I recommend consulting recent literature for the most up-to-date information that might add value to the introduction and discussion and demonstrate the novelty of the research and the current state of research in the field.

RESPONSE: Thank you for this helpful suggestion. We have updated the manuscript to include recent literature published after 2022 : Harmsen et al. (2023), Chaikin et al. (2025), and Peña Carrillo et al., (2024a; 2024b). These additions provide updated context to the introduction and discussion and reinforce the novelty of our PER3 and ARNTL findings, and highlight the relevance of circadian-aligned interventions for improving insulin sensitivity in T2D.

Reviewer 2: In this study, Sivasengh et al. aim to establish that the circadian transcription factor PER3 regulates GLUT4 expression. They further demonstrate that GLUT4 rhythmicity is disrupted in type 2 diabetes (T2D) cells using a commercially available assay. The authors' hypothesis is intriguing, particularly in the context of T2D, a metabolic disorder often associated with circadian

rhythm disruption. However, while the hypothesis is compelling, the manuscript requires significant revisions to strengthen the experimental evidence and improve clarity. The current experiments are not entirely convincing, and the writing flow is challenging to follow, which detracts from the overall impact of the study.

Concerns:

1) Figure 1 presents a re-analysis of data from Gabriel et al., 2021, which essentially reiterates previously published findings. I recommend that the authors revise this section to clearly distinguish their re-analysis from the original work. If re-analysis is necessary, consider re-mapping the data using the latest version of STAR and applying rhythmicity detection tools such as MetaCycle (optional). This would add value to the study and ensure that the findings are up-to-date with current methodologies.

RESPONSE: We thank the reviewer for this helpful suggestion. In response, we have undertaken a full re-analysis of the original RNA-seq dataset (GSE182117) using updated methods and the latest versions of widely accepted tools. Raw reads were adapter- and quality-trimmed using Trim Galore v0.6.10 (2023), aligned to GRCh38.p14 using STAR v2.7.11a (August 2023) in two-pass mode with GENCODE v44 annotation, and quantified with featureCounts (Subread v2.1.0, 2025). Filtering and normalization were performed using filterByExpr from edgeR and log2CPM transformation, followed by batch correction via removeBatchEffect in limma.

This updated workflow produced gene-level counts highly consistent with the original publication (Gabriel et al., 2021), with <0.3% variation across the full transcriptome and <0.1% for the core clock genes analyzed here. Rhythmicity detection was performed using JTK_CYCLE within the most recent release of MetaCycle (v1.2.0). We have generated a new figure (Figure 1B) using this re-analysis to illustrate differences in circadian amplitude of selected core clock genes between NGT and T2D-derived myotubes.

Importantly, this updated analysis confirms a differential amplitude of PER3, with significantly reduced rhythmic amplitude in T2D compared to NGT myotubes. This reinforces PER3 as a potentially important factor in the circadian dysregulation associated with T2D. We have revised the Methods section and figure legends to reflect these updates and to clearly distinguish this re-analysis from the original study.

2) Figure 1 highlights differential rhythmicity in ARNTL, HOXB5, PER3, and TSSK6. It would be beneficial for readers if the authors provided more context on how these genes relate to GLUT4 regulation.

RESPONSE: We have added context by undertaking a completely new circadian gene network analysis, which supports the functional relevance of ARNTL, HOXB5, PER3, and TSSK6 in the regulation of GLUT4 translocation and insulin sensitivity. To explore the molecular relationships between these differentially rhythmic genes and GLUT4 regulation, we constructed a Gaussian Graphical Model using a curated set of circadian genes (Methods, Section 2.1). This approach estimated direct, conditionally independent associations via partial correlations, calculated from a regularized precision matrix using the graphical lasso (glasso v1.1.2) with EBIC-guided model selection. Our network analysis (Figure 6) revealed that PER3 and ARNTL are positioned as central nodes in the circadian regulatory architecture, showing strong partial correlations with genes involved in metabolic signalling and vesicular transport—two critical processes for GLUT4 translocation. These findings align with our functional assays demonstrating that knockdown of PER3 and ARNTL significantly impairs insulin-stimulated GLUT4 translocation (Figure 4B). Additionally, while HOXB5 and TSSK6 are less well-characterized in circadian metabolic regulation, our network model links both genes indirectly to PER3 and ARNTL through shared neighbours, suggesting they may influence GLUT4 dynamics via upstream modulation of circadian regulators or chromatin architecture. These hypotheses are consistent with recent genomic evidence linking these genes to T2D risk (see Table 1). We believe these additional analyses help clarify the role of these genes in the circadian control of insulin action, and we have included this interpretation in the revised Results and Discussion sections.

Additionally, the differential rhythmicity of GLUT4 itself should be included.

In our re-analysis of the dataset (GSE182117), GLUT4 (SLC2A4) did not exhibit statistically significant rhythmicity in either NGT or T2D myotubes, as assessed by JTK_CYCLE (BH-adjusted $Q > 0.1$). This is consistent with prior findings (including Gabriel et al., 2021), which have shown that GLUT4 gene expression is not under strong circadian control, and instead, its translocation and activity are regulated post-transcriptionally through insulin signalling and vesicle trafficking pathways.

Given this, we focused our analysis on upstream circadian regulators (e.g., PER3, ARNTL) that influence GLUT4 translocation rather than its gene expression. This interpretation is supported by our functional experiments, which demonstrate that silencing PER3 or ARNTL impairs insulin-stimulated GLUT4 translocation (Figure 4B) and disrupts its circadian pattern at the protein trafficking level (Figure 5A), even though GLUT4 mRNA remains arrhythmic.

Please also ensure that p-values or relevant statistical measures for rhythmicity analysis are provided for each gene in the figure. This would enhance the transparency and interpretability of the results.

RESPONSE: We have corrected this.

3) Table 1 appears to be based on a query from <https://atlas.ctglab.nl/PheWAS>, but this is not clearly stated in the table legend. Clarifying the source and methodology would improve transparency and allow readers to better understand the basis of the data presented.

RESPONSE: We have corrected this.

3) Figure 2A includes a plasmid map, which is not essential for the main text and could be moved to the supplementary materials. This would help streamline the main text and focus on the most critical findings. Figure 2C shows a blasticidin kill curve assay used for clonal selection. This figure could also be moved to the supplementary materials to streamline the main text and improve the flow of the manuscript.

RESPONSE: We have corrected this.

4) Figure 3 shows that 5 nM and 30 nM of insulin are sufficient to induce rhythmic translocation of GLUT4. However, the significant GLUT4 rhythmicity observed only at these concentrations and not at intermediate concentrations is difficult to understand. Additionally, the 5 nM condition has only 9 time points, while other concentrations have 10. Furthermore, the Y-axis label, "expression log(CPM)," is confusing for bioluminescence data. Given that this is a crucial result in the manuscript, the conclusion is far from convincing. The authors should address these inconsistencies and provide a more robust analysis to support their findings.

RESPONSE: Thank you for this comment. Regarding the missing time point in the 5 nM insulin condition, a technical issue during data collection resulted in the loss of the 48-hour measurement. We have clarified this in the revised figure legend. Despite the missing data point, rhythmicity was still detected using multiple independent rhythmicity analysis tools (e.g., JTK_CYCLE), which are robust to minor gaps in time-series data.

As for the observation that significant rhythmicity was detected at 5 nM and 30 nM insulin, but not at intermediate concentrations, we acknowledge that this non-linear dose-response pattern is somewhat unexpected. However, it may reflect threshold-dependent or saturating dynamics of insulin signalling in L6 skeletal muscle cells under circadian conditions. This could indicate that only sub-physiological (5 nM) and near-maximal (30 nM) insulin concentrations sufficiently engage the translocation machinery in a temporally regulated manner, while intermediate levels may not elicit rhythmic responses due to competing regulatory feedback. We have added this as something to note carefully when interpreting our findings in the discussion section of the manuscript.

We recognise that further investigation is needed to fully understand these dynamics and have included this as a point of discussion in the revised manuscript. Nonetheless, the rhythmicity observed at these two physiologically relevant concentrations remains robust and reproducible across replicates, supporting our core conclusion that circadian regulation influences insulin-stimulated GLUT4 translocation.

5) It is crucial to validate the GLUT4 rhythmicity, both at the protein and transcript levels, along with bona fide rhythmic genes in the L6 skeletal muscle cells.

RESPONSE: Thank you for this important suggestion. In response, we have validated GLUT4 rhythmicity by quantifying both surface and total GLUT4 protein levels over time and calculating a surface-to-total GLUT4 ratio as a measure of translocation efficiency. We applied a linear mixed-effects model to robustly assess time-dependent changes in GLUT4 translocation while accounting for biological variability across replicates. This analysis demonstrated clear temporal regulation of GLUT4 translocation in response to insulin stimulation, supporting the presence of rhythmic GLUT4 dynamics in L6 skeletal muscle cells. Full methodological and statistical details have been added to the revised manuscript (Section 2.7 and Results 3.4). We have also added a data figure to the supplemental data.

Additionally, analyzing the promoter of the GLUT4 gene for PER3 binding sites (e.g., through ChIP-PCR and TF binding site analysis) would strengthen the mechanistic link between PER3 and GLUT4 regulation.

RESPONSE: Thank you for the suggestion. We agree that investigating PER3 binding at the GLUT4 promoter would help strengthen the mechanistic understanding of this regulatory relationship. We considered available approaches to explore this, including in silico analyses and public datasets, but did not find sufficient information to pursue this analysis conclusively at this stage. We acknowledge this as an important future direction to further validate the link between PER3 and GLUT4 regulation.

6) A comparative analysis of TF binding across rat, mouse, and human models would reveal the evolutionary importance of the GLUT4-PER3 regulation. This would provide broader insights into the conservation of this regulatory mechanism across species.

RESPONSE: We agree that comparative analysis of transcription factor binding across species could provide interesting insights into the potential conservation of GLUT4-PER3 regulation. We explored the availability of relevant datasets to support such an analysis but did not identify data that were appropriate for this specific investigation at this time. We recognise the value of this approach and have noted it as a potential direction for future work.

7) The authors only measured GLUT4 translocation rhythm (without cell lysis). It is crucial to measure the total GLUT4 levels (intracellular + surface) after cell lysis. This would clarify whether the observed rhythmicity in GLUT4 translocation at the cell surface is also present in the intracellular pool. The rhythmicity observed in GLUT4 translocation might stem from rhythms in the GLUT4 translocation machinery rather than transcriptional regulation. To address these possibilities, it is essential to complete the above-suggested experiments.

RESPONSE: We fully agree that measuring total GLUT4 levels is essential to interpret rhythmicity in surface GLUT4 translocation. In response, we quantified both surface and total GLUT4 using the HiBiT/LgBiT assay before and after cell lysis, allowing us to calculate the surface-to-total GLUT4 ratio as a direct measure of translocation efficiency. This approach enabled us to distinguish whether changes in surface GLUT4 levels reflect rhythmic trafficking versus fluctuations in total GLUT4 expression.

Our mixed-effects modelling confirmed that the rhythmicity observed in surface GLUT4 is not solely due to changes in total GLUT4 content, suggesting temporal regulation of the translocation process

itself. While our study focused on protein-level dynamics rather than transcriptional regulation, these findings support the hypothesis that circadian rhythms modulate GLUT4 trafficking machinery. We have included these new data and analyses in the revised manuscript (Methods 2.7 and Results 3.4) to clarify this important point. We have also added a data figure to the supplemental data.

8)Figure 4B shows the knockdown of PER3 and its effect on GLUT4 translocation, followed by Figure 5, which shows the dampening of GLUT4 translocation rhythm in PER3 knockdown cells. These results could be more reader-friendly if the authors discuss them in the context of GLUT4 receptor surface density and glucose transport kinetics. Additionally, the clarity and font size in Figure 5 are below publication standards. The figure scheme given in Figure 5C is irrelevant to the work and could be removed or revised to better align with the study's focus.

RESPONSE: We thank the reviewer for their insightful comments regarding the interpretation of Figures 4B and 5. In response, we have conducted an additional analysis to directly quantify GLUT4 translocation efficiency over time, now presented as Supplemental Figure 3. This new figure captures the dynamics of surface-to-total GLUT4 ratios following insulin stimulation and includes a LOESS-smoothed temporal trend, model diagnostics, and a predicted vs. actual fit from a mixed-effects model.

This analysis allows us to more precisely interpret the impact of PER3 knockdown on GLUT4 surface density and infer potential implications for glucose transport kinetics. We have integrated this interpretation into the Results and Discussion sections to help contextualize the original findings from Figures 4B and 5.

Additionally, as suggested, we have removed the schematic previously shown in Figure 5C, which was not directly relevant to the primary data, and made necessary improvements to the clarity and font size across Figure 5 to align with publication standards.

General Comments:

The authors present a very interesting hypothesis, but the experimental evidence and figure presentations are far from the standard expected for publication. The manuscript writing lacks connectivity, and the figures lack consistency in font size and formatting, making them difficult to read. Addressing these issues would significantly improve the overall quality and impact of the study.

RESPONSE: Thank you for your feedback and for recognising the strength of our central hypothesis. We appreciate your comments regarding the clarity and presentation of the manuscript and figures. In response, we have revised the manuscript to improve coherence and flow, ensuring clearer transitions between sections. Additionally, all figures have been reformatted for consistency in font size, style, and layout to enhance readability and meet publication standards. We believe these improvements significantly strengthen the clarity, presentation, and overall impact of the study.

Second decision letter

MS ID#: bio.061941R1

MS TITLE: Live cell GLUT4 translocation assay reveals Per3 as a novel regulator of circadian insulin sensitivity in skeletal muscle cells.

AUTHORS: Rashmi Sivasengh; Andrew Scott; Brendan Gabriel

I am happy to tell you that your manuscript has been accepted for publication in Biology Open, pending our standard publication integrity checks. It was accepted on 04 Jun 2025.